# Resting brain dynamics at different timescales capture distinct aspects of human behavior

Raphaël Liégeois [1,2,3], Jingwei Li [1], Ru Kong[1], Csaba Orban [1], Dimitri Van De Ville [2,3], Tian Ge[4,5], Mert R. Sabuncu[6] & B.T. Thomas Yeo[1,5,7,8]

Linking human behavior to resting-state brain function is a central question in systems neuroscience. In particular, the functional timescales at which different types of behavioral factors are encoded remain largely unexplored. The behavioral counterparts of static functional connectivity (FC), at the resolution of several minutes, have been studied but behavioral correlates of dynamic measures of FC at the resolution of a few seconds remain unclear. Here, using resting-state fMRI and 58 phenotypic measures from the Human Connectome Project, we find that dynamic FC captures task-based phenotypes (e.g., processing speed or fluid intelligence scores), whereas self-reported measures (e.g., loneliness or life satisfaction) are equally well explained by static and dynamic FC. Furthermore, behaviorally relevant dynamic FC emerges from the interconnections across all resting-state networks, rather than within or between pairs of networks. Our findings shed new light on the timescales of cognitive processes involved in distinct facets of behavior.

[1] Department of Electrical and Computer Engineering, Clinical Imaging Research Centre, N.1 Institute for Health and Memory Networks Program, National University of Singapore, Singapore 117583, Singapore. [2] Institute of Bioengineering, Centre for Neuroprosthetics, École Polytechnique Fédérale de Lausanne, 1015 Lausanne, Switzerland. [3] Department of Radiology and Medical Informatics, University of Geneva, 1205 Geneva, Switzerland. [4] Psychiatric & Neurodevelopmental Genetics Unit, Center for Genomic Medicine, Massachusetts General Hospital, Boston, MA 02114, USA. [5] Athinoula A. Martinos Center for Biomedical Imaging, Massachusetts General Hospital, Charlestown, MA 02129, USA. [6] School of Electrical and Computer Engineering, Cornell University, Ithaca, NY 14853, USA. [7] Centre for Cognitive Neuroscience, Duke-NUS Medical School, Singapore 169857, Singapore. [8] NUS Graduate School for Integrative Sciences and Engineering, National University of Singapore, Singapore 119077, Singapore. Correspondence and requests for materials should be addressed to R.L. (email: Raphael.Liegeois@epfl.ch) or to B.T.T.Y. (email: Thomas.Yeo@nus.edu.sg)

Brain activity is highly organized in space and in time, even in resting-state conditions[1]. This intrinsic organization, classically evaluated from resting-state functional connectivity[2] (FC), has been shown to encode various behavioral aspects such as integration of cognition and emotions[3], monitoring of external environment[4], intellectual performance[5,6], and emergence of stimulus-independent thoughts[7]. FC has also been also used as a neuroimaging marker of several pathologies, including Alzheimer's disease[8,9], major depressive disorders[10], Parkinson's disease[11], schizophrenia[12], and autism[13]. More recently, the advent of large neuroimaging and behavioral datasets has allowed the further exploration of the FC behavioral counterparts, showing intricate contributions of cognitive, emotional, social, and demographic aspects[14].

Importantly, all these studies use static measures of FC, which reflect the average functional organization of entire neuroimaging recordings typically running over several minutes. However, there is recent converging evidence, suggesting that the resting brain navigates through different functional connectivity configurations at much faster timescales on the order of seconds[15,16]. Therefore, new dynamic measures of resting-state FC exploiting these faster changes have been proposed[17] and their behavioral counterparts have in turn been explored, showing links to cognitive flexibility[18], drug use[19], and mind-wandering[20]. Yet, the comparison of static and dynamic measures of FC has only been proposed in specific applications such as temporal lobe epilepsy[21] or the description of eating behaviors[22], and a comprehensive analysis exploring which types of behavioral features emerge from functional interactions at different timescales is missing.

We explore this question using a discovery dataset ($N = 419$) and a replication dataset ($N = 328$), comprising high-resolution resting-state functional magnetic resonance imaging (fMRI) time series and 58 behavioral measures spanning cognitive, emotional, social, and personality traits from the Human Connectome Project[23] (HCP). First, we compare the extent to which static and dynamic FC capture behavioral information. We then investigate whether the behavioral relevance of FC markers is preferentially encoded in within- or between- network connectivity. Finally, we test if static and dynamic FC capture complementary behavioral information. FC dynamics are evaluated using a first-order autoregressive (AR-1) model of resting-state fMRI data[24,25]. AR-1 models exploit the temporal ordering of fMRI time series to capture dynamic FC happening at a resolution of a few seconds to which static approaches are blind[16], without suffering from the limitations of classical sliding window methods[26,27]. FC dynamics are then linked to the 58 behavioral measures using a variance component model[28]. This model has been extensively used in genome-wide complex trait analyses and was recently applied to study the neuroanatomical signatures of traits such as cognitive or clinical measures[29].

We find that FC dynamics specifically encode behavioral measures evaluating performance in tasks, whereas self-reported measures are explained equally well by static and dynamic FC. We argue that this reflects the nature of the functional processes involved in the corresponding behavioral experiments. On the one hand, task-based metrics engage cognitive processes at timescales on the order of a few seconds that can be captured by FC dynamics. On the other hand, self-reported measures might reflect trait-like properties that are less likely to change over a few seconds, therefore being equally well explained by functional patterns averaged over longer periods as encoded in static FC. Furthermore, our results also suggest that task-performance scores are defined by whole-brain FC dynamics involving the interaction between multiple resting-state networks. Overall, more than providing a mere statistical marker of task-performance, these findings offer new insights into the timescales of the

cognitive processes involved in the execution of various tasks[30–32], thereby also supporting the 'networked-brain" paradigm that has emerged in recent years[33].

## Results

**Behavioral counterparts of static and dynamic FC**. We used data from 419 unrelated HCP subjects[34] to explore the extent to which behavioral information is encoded in dynamic markers of resting-state functional connectivity (FC), beyond classical static measures of FC. We selected 58 behavioral measures from the HCP dataset covering cognitive, social, emotion, and personality traits (see Supplementary Table 2) from which age, gender, race, education, and motion (mean FD) were regressed.

FC markers were estimated from the HCP resting-state fMRI dataset. Classical preprocessing was performed, followed by a parcellation into 400 cortical regions of interest (ROIs) and 19 subcortical ROIs[35]. Subject-specific static FC markers were computed by averaging correlation matrices of fMRI time series across runs. Dynamic FC markers were defined from an AR-1 model identified from the concatenation of the runs for each subject (Methods). We chose to represent FC dynamics using an AR-1 model for several reasons. First, we have shown recently that AR-1 models, by exploiting the statistical link between successive time points, capture FC dynamics significantly better than a hidden Markov model explicitly representing switches between different states with an equivalent number of parameters[16]. Second, the hierarchical organization of brain network dynamics was found to be reproduced by an AR-1 model of fMRI time series[36]. Finally, lag threads, which also exploit the sequential ordering information of time series (although they focus on identifying temporal sequences of propagated activity rather than connectivity patterns) were shown to provide meaningful markers of intrinsic brain function[37].

The link between FC markers and behavioral measures was studied using a variance component model[28,38]. The model inputs are (i) a matrix containing the 58 behavioral measures for the $N = 419$ subjects and (ii) at least one $N \times N$ matrix, called a similarity matrix and denoted by $K$, whose $i,j$-th entry encodes the similarity between (static or dynamic) FC of subjects $i$ and $j$. Note that static FC matrices are symmetric, whereas dynamic FC matrices are non-symmetric. The model estimates the level of behavioral variability that is explained by FC variability, both on average over all behavioral measures, as well as for each behavioral measure (Methods[38]).

**Dynamic FC markers encode more behavioral information**. We first compared the level of behavioral variance explained by static and dynamic FC markers. To this end, we ran the multivariate variance component model twice: once using a similarity matrix encoding the inter-subject similarity of static FC patterns, and once using similarity of dynamic FC patterns.

Figure 1a shows that on average over the 58 behavioral measures, dynamic FC markers capture more behavioral variance than static FC ($p = 8.31 \times 10^{-4}$; two-tailed $t$-test), and Fig. 1b presents the results for eight individual phenotypic measures. Results for the 50 remaining HCP measures are found in Supplementary Fig. 1.

**Dynamic FC specifically encodes task-based measures**. Even if dynamic FC encodes more behavioral information than static FC on average, results of Fig. 1b show that some behavioral measures are not better explained by dynamic FC (e.g., Meaning of Life, Loneliness or Perceived Stress). In order to explore whether FC dynamics specifically capture certain types of behavioral measures, we ranked the 58 HCP measures based on the extent to

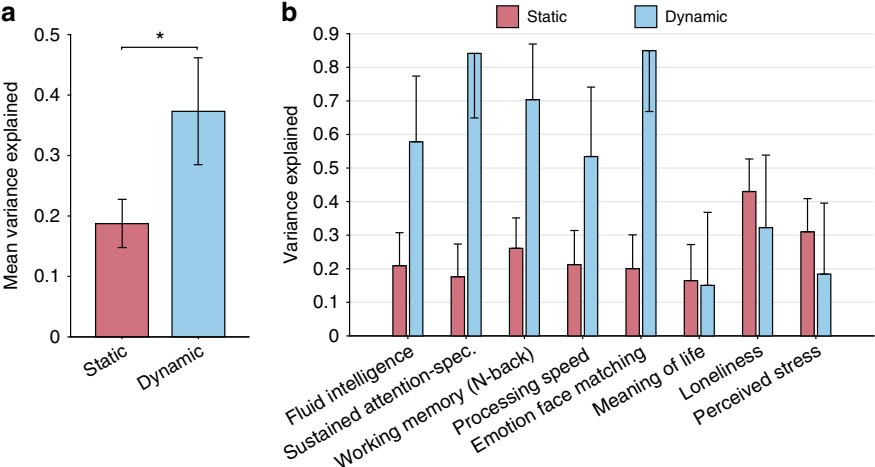

**Fig. 1** Dynamic FC explains more behavioral variance than static FC. **a** On average over 58 behavioral measures, dynamic FC (blue, 37%) explains more behavioral variance than static FC (red, 19%) ($p = 8.31 \times 10^{-4}$; two-tailed $t$-test). **b** Variance explained for eight representative measures. Here, static FC utilizes Pearson's correlation, while dynamic FC utilizes the coefficient matrix of a first-order autoregressive model. Error bars indicate standard deviation (SD) of the estimates

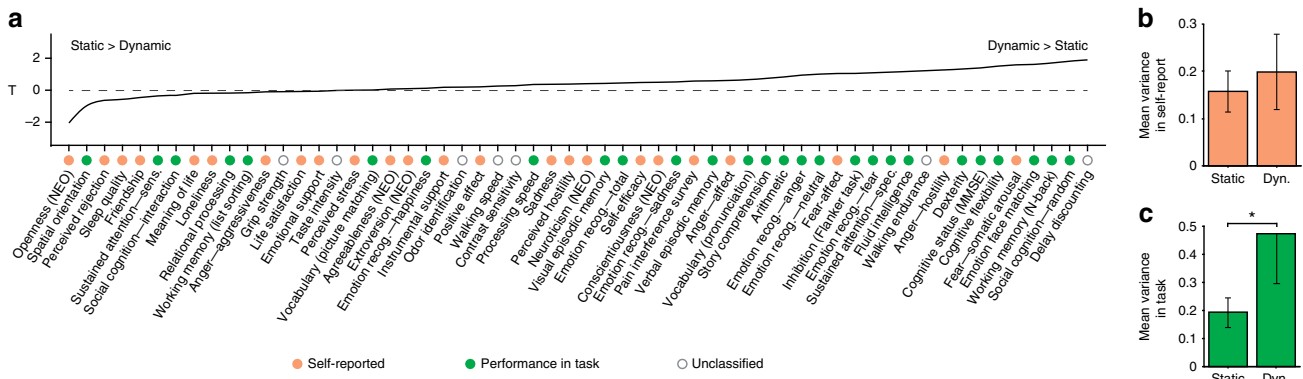

**Fig. 2** Dynamic FC explains larger behavioral variance than static FC in task-performance measures. **a** Behavioral measures are ordered based on whether dynamic FC explains more variance than static FC. A positive $t$-statistic T suggests that dynamic FC explains more variance than static FC. Behavioral measures corresponding to task-performance are marked with a green dot and self-reported measures are marked with an orange dot. **b** No statistically significant difference ($p > 0.10$: two-tailed $t$-test) was found in the mean variance explained by static and dynamic FC in self-reported measures. **c** Measures of performance in task are on average significantly better explained ($p = 1.75 \times 10^{-3}$; two-tailed $t$-test) by dynamic FC. Error bars indicate SD of the estimates

which dynamic FC better explain their variability, as compared to static FC. To this end, we repeated the procedure for the 58 measures and computed 58 $t$-statistics, denoted by $T$, of the difference between behavioral variance explained by static and dynamic FC for each measure (Supplementary Methods). Negative values of $t$-statistics indicate that static FC tends to better explain the measure, whereas behavioral measures with positive statistics are better explained by dynamic FC, as indicated in Fig. 2a.

This ranking seems to draw a dichotomy between "task-performance" and "self-reported" measures. On the one hand, the first category includes metrics that use participant's performance in a task to assess a trait (e.g., working memory, spatial orientation) and are marked with green dots in Fig. 2a. On the other hand, "self-reported" measures (orange dots in Fig. 2a) rely on subjective appraisal of traits (e.g., loneliness, life satisfaction). No label was attached to the measures with no clear classification in one of these categories. We find that dynamic FC better

explained task-performance measures ($p = 1.75 \times 10^{-3}$, Fig. 2c), whereas no statistically significant difference could be found in the capacity of both markers to explain self-reported measures (Fig. 2b). We also find that the difference of the differences between static and dynamic explained variances observed in Fig. 2b, c is itself different from zero ($p = 3.62 \times 10^{-3}$; two-tailed $t$-test). This interaction effect confirms that the difference observed in Fig. 2c is related to the task condition and not only driven by the main effect shown in Fig. 1a. Moreover, the result of Fig. 2c is reproduced using subcategories of task-based measures (Supplementary Fig. 3). Overall, the better average capacity of dynamic FC to explain behavioral measures seems to be driven by its increased capacity to explain task-based measures.

**Behavior-related FC dynamics arise from network interactions.** Functional interactions between brain networks have been shown to play a key role during the execution of tasks[39] and in the description of traits[40]. We tested whether interaction between

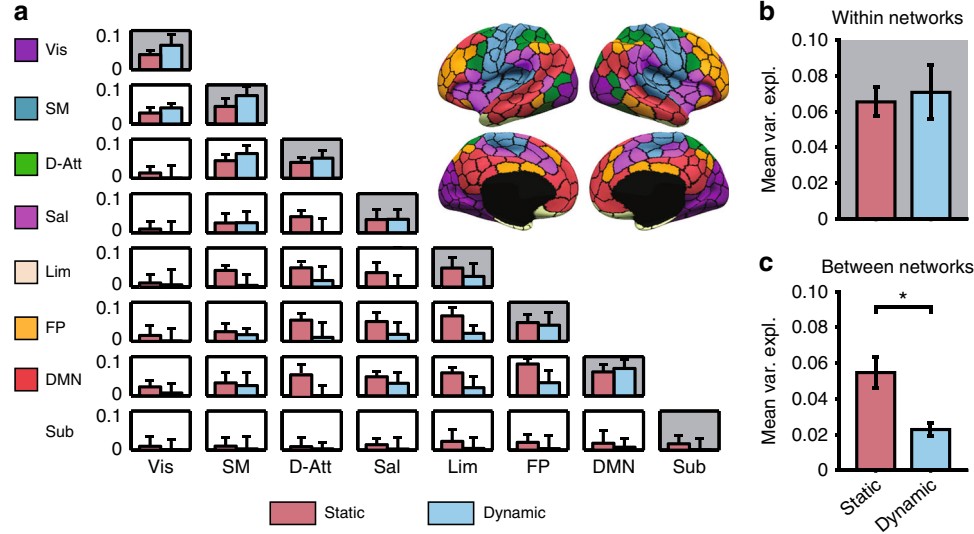

**Fig. 3** Dynamic FC does not explain more behavioral variance than static FC within (pairs of) networks. **a** Behavioral variance explained by within-network (shaded diagrams) and between-network (unshaded diagrams), network static and dynamic FC. Seven cortical networks were used: visual (VIS), somatomotor (SM), dorsal attention (D-Att), salience (Sal), limbic (Lim), frontoparietal (FP), default mode network (DMN) and we also gathered the 19 subcortical areas (Sub). **b** There is no statistically significant difference in behavioral variance explained by within-network static and dynamic FC. **c** Between-network static FC explains more behavioral variance than between-network dynamic FC ($p = 8.31 \times 10^{-3}$; two-tailed $t$-test). Error bars indicate SD of the estimates

resting-state networks were also critical for extracting behavioral information from FC. To this end, the same model as described above was used but similarity matrices were not computed from the whole static or dynamic FC matrices. Instead, only sub-blocks of the FC matrices corresponding to (pairs of) well-known resting-state networks were used. In other words, we tested how behavioral variability is encoded in the variability of (pairs of) resting-state networks connectivity patterns. We used a common partition in seven cortical resting-state networks[41] and included subcortical areas (Methods), as shown in Fig. 3.

The average behavioral variance explained by static and dynamic FC restricted to within or between networks is under 10% for almost all the pairs of networks. Not surprisingly this is lower than the behavioral variance explained from the whole-brain connectivity patterns (19% for static FC and 37% for dynamic FC), as anticipated from previous findings showing that individual FC fingerprinting is distributed throughout the brain[42]. More unexpected is the fact that FC dynamics do not seem to carry more behavioral information than static FC. On the contrary, on average over all inter-network connections (Fig. 3, unshaded diagrams), static FC explained more behavioral variance than dynamic FC ($p = 8.31 \times 10^{-3}$; two-tailed $t$-test), whereas no statistically significant difference was found for within-networks connections (Fig. 3, shaded diagrams).

**Testing complementarity between static and dynamic FC**. We have shown that on average FC dynamics encode more behavioral information than static FC (Fig. 1), especially for task-performance measures (Fig. 2). However, this does not mean that static FC is not capturing any additional behavioral information not encoded by dynamic FC. To test this, we used a generalized version of the multivariate variance component model that takes multiple similarity matrices -in our case two: the ones computed from static and dynamic FC- as inputs and estimates the level of behavioral variance explained by the combination of these similarity matrices (Supplementary Methods).

The average behavioral variance explained by combining static and dynamic FC is shown in dark blue in Fig. 4a. Results for eight representative measures are shown in Fig. 4b and results for the 50 remaining traits are found in Supplementary Fig. 2. In Fig. 4a, the combined value is significantly higher than static FC ($p = 4.73 \times 10^{-4}$; two-tailed $t$-test), confirming the fact that FC dynamics contains information above and beyond static FC. However, no statistical difference was found between average combined results and dynamic FC ($p > 0.10$, see Supplementary Table 1 for details), which suggests that the information encoded by static FC is largely encoded in dynamic FC.

**Dynamic FC interactions driving task-performance**. We now explore which dynamic FC interactions contribute to the overall association with task-performance (Fig. 2c). We used a reformulation of the variance component model defined in Eq. (2) that revealed the relative contribution of the interaction between each pair of (sub)networks to the overall explained variance (Supplementary Eq. (10)). The results are shown in Fig. 5. It can be seen that default C and frontoparietal C, together with the subcortical regions, are contributing the most to the association between dynamic FC and task-performance.

**Replication dataset**. The findings shown in Figs. 1–4 were replicated in a second group of 328 unrelated HCP subjects. More precisely, all significant differences found in Figs. 1–4 were also found to be significant in the replication dataset (more details are found in Supplementary Figs. 5–9 and Supplementary Table 1). The replication dataset was composed of the second subject of each HCP family containing more than one person. We note that it is therefore not completely independent from the discovery dataset.

**Additional control analyses**. We performed a series of control analyses to evaluate the impact of various processing steps in our baseline analysis. More specifically, we tested the impact of (i) including the variance of the mean cortical grayordinate signal as

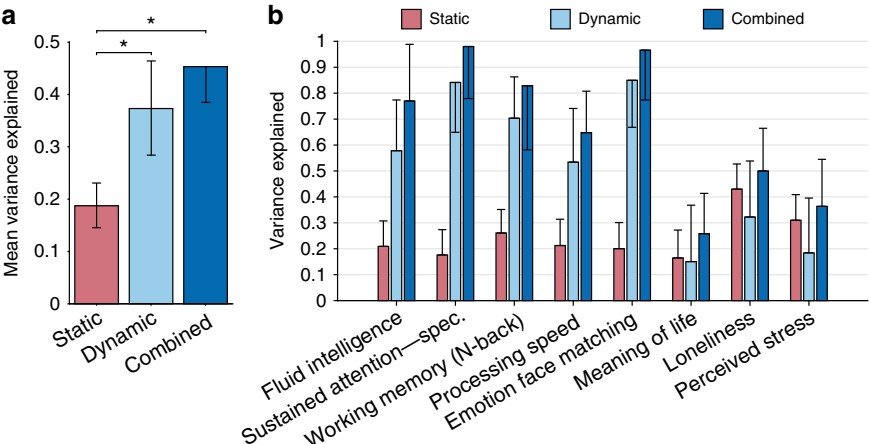

**Fig. 4** Combined static and dynamic FC does not capture more behavioral variance than dynamic FC alone. **a** Average variance explained across 58 behavioral measures using static FC (red), dynamic FC (light blue), and the combination of these two (dark blue). **b** Variance explained for eight representative measures. Error bars indicate SD of the estimates

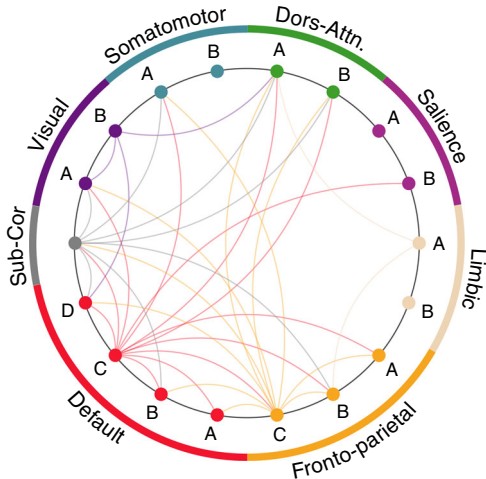

**Fig. 5** Dynamic FC interactions contributing the most to the association with task-performance. Networks and corresponding colors are the same as in Fig. 3, and subnetworks are defined following the 17-network parcellation of Schaefer et al.[35], as reported in Supplementary Fig. 4. The colors of the edges are defined by their destination and only connections surviving an FDR correction at the level $q = 0.05$ are shown

a covariate in the variance component model, (ii) evaluating the static and dynamic FC matrices from fMRI time series from which the mean cortical grayordinate signal was not regressed, (iii) including head motion metrics as covariates in the variance component model, (iv) evaluating the static and dynamic FC matrices from full (i.e., uncensored) fMRI time series, (v) the number of behavioral measures considered in the variance component model, and (vi) the relative contributions of static and dynamic FC to the overall variance explained within the combined variance component model. The variance component model appeared to be robust to these changes and in each case, our main findings were reproduced (Supplementary Figs. 10–12).

## Discussion

Exploring how resting-state functional organization is linked to various behavioral traits is a central neuroimaging research question. This organization is classically evaluated from functional connectivity (FC), and recent evidence has shown that exploiting the dynamical properties of FC instead of the classical static FC metrics could open new avenues to interpret brain functioning at different timescales. In this study, we aim at refining our understanding of the behavioral information carried by FC dynamics. To this end, we explored the extent to which resting-state static and dynamic FC measures relate to a large repertoire of measures covering cognitive, social, emotion, and personality traits. We first show that, on average over 58 selected behavioral measures, FC dynamics encode significantly more behavioral information than a common static FC metric. This confirms current findings that have highlighted the advantage of resting-state dynamic FC measures over their static counterparts in describing mindfulness[43], disease[21], and eating behaviors[22].

Interestingly, FC dynamics within well-known resting-state networks, or between pairs of networks, did not capture more behavioral information than static FC in the same networks (Fig. 3). For example, static inter-network FC was shown to explain more behavioral variance than dynamic inter-network FC on average. These results might seem counter-intuitive at first sight and suggest that similarity measures derived from local patterns of FC do not complement each other in the same way in the static and dynamic cases. In other words, the advantage of dynamic FC in explaining the behavioral information observed in Fig. 1 is encoded in the global dynamic FC interaction patterns. From a methodological point of view, this also indicates that even if dynamic FC uses richer statistical information than static FC by relaxing the static assumption associated to this metric, dynamic FC should not a priori and automatically be considered as a better neuroimaging marker than static FC.

In Fig. 2, we show that FC dynamics specifically encode measures of performance in a task, such as working memory tasks, whereas static and dynamic FC explain self-reported measures, such as the perception of loneliness, equally well. This additional information is found to be encoded in the global dynamic FC patterns, and not confined to single areas or networks (Fig. 3). More precisely, Fig. 5 suggests that the default mode and frontoparietal networks drive the integration of the dynamic FC coming from other networks. This is in line with previous findings identifying these areas as hubs of the dynamic functional connectome[44], and further supports the importance of coupled default network and frontoparietal activities during task-performance[45].

Taken together, these results support the line of current findings indicating that different phenotypic measures such as pain[40], perception[46], and vigilance[47] are encoded in dynamic interactions between multiple areas and not within single networks. Then, even if they concern interactions between resting-state networks, our results interestingly echo the nature of network interactions involved in task-based conditions. Indeed, the execution of tasks relies on a coordinated activation of different networks at faster timescales that can be captured by dynamic FC[31,32], and to which static measures of FC are blind. On the contrary, self-reported measures could be considered as trait-like properties that might therefore be explained equally well by average FC patterns encoded in static FC. Converging results also suggest that simple tasks exhibit segregated activation patterns whereas complex tasks involving multiple cognitive processes (e.g., working memory or visuospatial attention) require an integrated activation of multiple intrinsic networks[39,48,49] and activate flexible brain regions such as connector hubs[50,51]. It does not seem unreasonable to assume that task-performance measures, in which subjects are incited to reach a high score in a test that often involves multiple or coordinated actions, capture more complex behavioral traits -as defined above- than self-reported measures. More generally, these results constitute empirical evidence supporting the 'networked-brain' paradigm that has emerged in recent years. This paradigm essentially views the brain as a multiscale network producing complex spatio-temporal activity patterns rather than an ensemble of neuronal populations with localized specification[33,52,53].

The fact that FC dynamics seem to capture more complex network interactions than static FC is also supported by the links that have been drawn between FC and the underlying brain anatomy, or structural connectivity (SC). As static FC is on average closer to SC, the dynamic FC repertoire also captures excursions from SC that are characterized by higher efficiency and lower modularity[54]. These fluctuations of modular organization, encoded in FC dynamics, were shown to operate at different timescales[30] and to support the periodic (de)coupling of resting-state networks[55], possibly constituting a signature of consciousness[56] and allowing for a more efficient transfer of neuronal information[57].

Other converging findings suggest that the amplitude of FC dynamics variability decreases during task as compared to resting-state[58,59]. Altogether with the results of Figs. 2 and 3 indicating that task-based behavioral measures are specifically encoded by resting-state FC dynamics, we might hypothesize that there exists a resting-state 'dynamic reservoir" that is recruited when leaving rest and, which defines task-performance. This dynamic reservoir, encoded by FC dynamics and not by simpler static FC markers as shown in Fig. 4, emerges from highly integrated connections involving multiple intrinsic networks. As such, we could also interpret this dynamic connectivity structure as a signature of the human connectome evolution that tends to attain efficient organizations to perform complex tasks[60].

The dichotomy of behavioral measures proposed in Fig. 2 is motivated by the statistical difference between static and dynamic FC that seems to capture distinct behavioral properties. This dichotomy presents some limitations. First, the classification of behavioral measures in one of the two proposed categories, 'task-performance" or 'self-reported", was not always straightforward. For example, the Delay Discounting task measure was left unclassified as one could argue that it is not capturing a performance, associated with an underlying truth or optimal score, in the same way a classical task such as the Working Memory task is. For the same reason, other measures (grip strength, odor identification, walking speed, contrast sensitivity, taste intensity, and walking endurance) were not classified (Supplementary

Table 2). However, we note that classifying these measures in one or the other category did not significantly change the results in Fig. 2. That said, considering only two categories of behavioral measures disregards the multifactorial nature of behavior and the repertoire of behavioral measures could be approached using other classification criteria such as trait vs. state[61] or intrinsic vs. extrinsic[62].

We referred to the AR model of BOLD time series as 'dynamic" following the systems theory literature[63]. This nomenclature is motivated by the fact that such models, by accounting for the memory present in the time series (i.e., $x_t$ depends on $x_{t-1}$), are able to reproduce empirical fluctuations in the multivariate time series of interest much better than memoryless (or 'static", following the same nomenclature) models[16]. The AR model is also used to compute the dominant dynamic modes shaping resting-state brain function[64]. Overall, this model can be seen as a compact way to summarize the temporal fluctuations of BOLD and FC time series that are directly exploited by time-varying models[17,65].

In summary, static measures of FC provide a measure of brain function averaged over several minutes. This is an oversimplification and new dynamic measures capturing the temporal changes of brain function on the order of a few seconds have been proposed. While these new measures were shown to capture more statistical properties of fMRI data, their behavioral relevance above and beyond static FC remains unclear. Here, we have shown using 747 HCP subjects and 58 behavioral measures that FC dynamics specifically capture measures of performance in tasks by leveraging the dynamic information encoded in multiple-network interactions. On the contrary, self-reported measures are equally well explained by static and dynamic measures of FC. Overall, we believe our work opens up future possibilities to a better characterization of the cognitive processes shaping the various facets of human behavior.

## Methods

**Data and preprocessing**. We used data of the HCP 1200-subjects release comprising structural MRI, resting-state functional MRI, and behavioral measures of young (ages 22–35) and healthy participants drawn from a population of siblings[23]. All imaging data were acquired on a 3-T Siemens Skyra scanner using a multi-band sequence. Functional images have a temporal resolution of 0.72 s and a 2-mm isotropic spatial resolution whereas structural images are 0.7-mm isotropic. For each subject, four 14.4 min runs (1200 frames) of functional time series were acquired[34]. Resting-state fMRI data was projected to the fs_LR surface space using the multimodal surface matching method (MSM-All)[23,66]. Both cortical and sub-cortical data were cleaned using the ICA-FIX method[67,68] and saved in CIFTI grayordinate format. This cleaning procedure included the regression of 24 motion-related parameters (six classical motion parameters, their derivatives, and the squares of these 12 parameters). Motion censoring was then applied by removing frames with FD > 0.2 mm or DVARS > 75, as well as one frame before and two frames after these frames[69,70]. Remaining segments containing less than five frames were also removed and runs with >50% of censored frames were discarded. Linear trends and mean cortical grayordinate signal were regressed and censored frames were ignored to compute regression coefficients. Mean cortical grayordinate signal regression was performed because this step was shown recently to strengthen the association between FC metrics and behavioral measures[71]. However, we note that not regressing mean cortical grayordinate signal yielded similar conclusions (Supplementary Fig. 10). Finally, fMRI time series were parcellated into 419 regions of interest (ROIs) comprising 400 cortical areas[35] and 19 subcortical areas defined in Freesurfer. Static functional connectivity was obtained for each subject based on the pearson's correlation matrices, computed from uncensored frames for each run, which were Fisher z-transformed, averaged over runs and transformed back to r-space. Dynamic functional connectivity measures were estimated from the model parameter of a first-order autoregressive representation of fMRI time series:

$$x_t = A \cdot x_{t-1} + \epsilon_t \qquad (1)$$

where $x_t \in \mathcal{R}^{N_R \times 1}$ represents the fMRI time series in the $N_R = 419$ ROIs at time $t$, $A \in \mathcal{R}^{N_R \times N_R}$ is the model parameter that encodes the linear relationship between successive time points, and $\varepsilon_t \in \mathcal{R}^{N_R \times 1}$ are the residuals of the model[16]. The model parameter $A$ was identified from the concatenation of the uncensored sections of the different runs, while ignoring transitions between uncensored sections and

transitions between runs. The proportion of variance explained by this model, defined as $R^2 = 1 - \frac{||\mathrm{Cov}(\epsilon_t)||_F}{||\mathrm{Cov}(x_t)||_F}$ analogous to the definition of $R^2$ for univariate AR models, and where $||\cdot||_F$ denotes the Frobenius norm[72], is 69.3% ± 11.2% (computed over all subjects, including the replication dataset).

We selected 58 behavioral measures, including cognitive, social, emotion and personality traits (Supplementary Table 2). These measures consist of metrics from the NIH Toolbox and some well-known non-NIH measures (e.g., NEO-FFI). Details about behavioral measures can be found in HCP S1200 Data Dictionary and Barch et al.[73]. These measures were classified either as 'task-performance", 'self-reported", or 'unclassified" if no clear belonging to one of the first two categories applied (Supplementary Table 2).

Among the HCP 1200-subjects release, 1029 subjects had at least one run that was not discarded after applying the preprocessing rules. Excluding the subjects with missing or problematic entries for some behavioral measures further reduced the dataset to 953 subjects belonging to 419 families. To avoid the influence of shared genetic and environmental factors, we kept the first subject from each family leading to a final set of $N = 419$ unrelated subjects. Among the 419 families used for the initial set, 91 'families' contained only one subject and hence these families were discarded in the construction of the replication dataset that contained 328 subjects.

**Variance component model**. We regressed age, gender, race, education, and motion (mean FD) from the 58 phenotypic measures, which were then quantile normalized. We used the multivariate variance component model developed by Ge et al.[38] to link FC markers and behavioral measures:

$$Y = C + E \qquad (2)$$

where $Y$, $C$, and $E$ are $419 \times 58$ matrices. $Y$ contains the 58 processed behavioral measures for all 419 subjects. $\mathrm{Vec}(C) \sim \mathcal{N}(0, \Sigma_c \otimes F)$ and $\mathrm{Vec}(E) \sim \mathcal{N}(0, \Sigma_e \otimes I)$, where $\mathrm{Vec}(.)$ is the matrix vectorization operator, $\otimes$ is the Kronecker product of matrices, and $I$ is the identity matrix. $F$ is a similarity matrix such that $F(i, j)$ encodes the (static or dynamic) FC similarity between subjects $i$ and $j$, and is defined as the correlation between the static FC (or dynamic FC) matrices of the two subjects. $\Sigma_c$ and $\Sigma_e$ are unknown $58 \times 58$ matrices to be estimated from $F$ and $Y$. The variance explained by (static or dynamic) FC markers, denoted by $M$, is computed as:

$$M = \frac{\mathrm{Tr}(\Sigma_c)}{\mathrm{Tr}(\Sigma_c) + \mathrm{Tr}(\Sigma_e)}, \qquad (3)$$

where $\mathrm{Tr}(.)$ is the trace operator. $M$ measures how much inter-subject behavioral variability is explained by inter-subject FC variability[38]. Equation (3) is computed twice, once using static FC to build the FC similarity matrix in (2) and once using dynamic FC to build this matrix. $M_i$ for a single behavioral measure is given by $M_i = \Sigma_c(i, i)/(\Sigma_c(i, i) + \Sigma_e(i, i))$. We used the variance component model because it provides an easy way to summarize high-dimensional FC patterns and produces one scalar (variance explained) for each behavioral measure that is normalized (between 0 and 1) and directly comparable across measures and processing methods.

**Complexity of static and dynamic variance component models**. Static FC similarity matrices are built from the correlation matrix of resting-state fMRI time series. These correlation matrices are symmetric and of size $N_{\mathrm{ROI}} \times N_{\mathrm{ROI}}$, where $N_{\mathrm{ROI}}$ is the number of regions of interest, and hence their degrees of freedom (d.o.f.) is $N_{\mathrm{ROI}} \times (N_{\mathrm{ROI}} - 1)/2$. On the other hand, dynamic FC is built from the non-symmetric coefficient matrix of a first-order AR model of size $N_{\mathrm{ROI}} \times N_{\mathrm{ROI}}$, and hence their d.o.f. is $N_{\mathrm{ROI}}^2$. It should be noted, however, that this difference in d. o.f. is not present in the static and dynamic variance component models as the static and dynamic FC matrices are only used in intermediate steps of the computation of the similarity matrices of the variance component model. These similarity matrices are in both the static and dynamic case symmetric and of size $N$ with $N \times (N - 1)/2$ degrees of freedom, where $N$ is the number of subjects. Therefore, there is no complexity bias giving an advantage to one or the other approach.

**Statistic inference**. We use the delete-1 Jackknife approach[74] to evaluate the statistical difference between the variance explained by two different methods (e.g., when $M$ is computed using only static FC vs. only dynamic FC). Let $\hat{M}^A$ and $\hat{M}^B$ be the estimates of behavioral variance explained using the methods A and B to be compared. Denoting by $\hat{M}_{-i}$ the estimates computed from the whole dataset without subject $i$, the delete-1 Jackknife estimate of the mean difference between methods A and B is:

$$\hat{M}_{\mathrm{jack}} = \frac{1}{N} \sum_{i=1}^{N} \left( \hat{M}_{-i}^A - \hat{M}_{-i}^B \right) = \frac{1}{N} \sum_{i=1}^{N} \hat{M}_{-i}, \qquad (4)$$

where $N = 419$ is the number of subjects, and $\hat{M}_{-i} = \hat{M}_{-i}^A - \hat{M}_{-i}^B$ is the difference of explained variance for the $i$-th Jackknife sample. The variance estimate of the

mean difference of explained variance is:

$$\hat{V}_{\mathrm{jack}} = \frac{N-1}{N} \sum_{i=1}^{N} \left( \hat{M}_{-i} - \hat{M}_{\mathrm{jack}} \right)^2. \qquad (5)$$

With a large sample size ($N = 419$), the estimator $\left( \hat{M}_{\mathrm{jack}} - M \right)/\sqrt{\hat{V}_{\mathrm{jack}}}$ is assumed to follow a standard normal distribution under the null hypothesis ($M = 0$) and a two-tailed p-value can be computed. All significant results survived FDR correction at $q < 0.05$ (see Supplementary Table 1 for details).

**Reporting summary**. Further information on research design is available in the Nature Research Reporting Summary linked to this article.

## Data availability

The HCP data is publicly available at http://www.humanconnectomeproject.org/data/; informed consent was obtained from all HCP participants[23].

## Code availability

All code is publicly available at https://github.com/RaphaelLiegeois/FC-Behavior/.

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

## Acknowledgements

We would like to thank Russ Poldrack, Timothy Laumann, and Jean-Baptiste Poline for their insightful feedback. This research was supported in parts by the CHIST-ERA IVAN project (20CH21 174081), the Center for Biomedical Imaging (CIBM) of the Geneva-Lausanne Universities and the EPFL, the Singapore MOE Tier 2 (MOE2014-T2-2-016), NUS Strategic Research (DPRT/944/09/14), NUS SOM Aspiration Fund (R185000271720), Singapore NMRC (CBRG/ 0088/2015), NUS YIA, Singapore NRF fellowship (NRF-NRFF2017-06), NIH grant K99AG054573, and the Neuroimaging Informatics and Analysis Center (1P30NS098577). The computational work was partially performed on resources of the National Supercomputing Center, Singapore https://www.nscc.sg. Data were provided by the Human Connectome Project, WU-Minn Consortium (Principal Investigators: David Van Essen and Kamil Ugurbil; 1U54MH091- 657) funded by the 16 NIH Institutes and Centers that support the NIH Blueprint for Neuroscience Research; and by the McDonnell Center for Systems Neuroscience at Washington University.

## Author contributions

R.L., C.O., T.G., M.R.S., and B.T.T.Y. designed research; R.L., J.L, and R.K. performed research; R.L., J.L., R.K., and T.G. contributed new reagents/analytic tools; R.L., J.L., T.G., and B.T.T.Y. analyzed data; and R.L., D.V.D.V., T.G., M.R.S., and B.T.T.Y. wrote the paper.

## Additional information

**Competing interests:** The authors declare no competing interests.

