## [Peer Review File · Nature Communications]

Reviewers' comments:

Reviewer #1 (Remarks to the Author):

"Resting brain dynamics at different timescales," by Liegeois et al.

This paper compares the behavioural relevance of zero lag cross correlations in resting state BOLD data to 1-lag correlations derived from a first order autoregressive model.

The study hence combines prevailing interest in the phenotypic correlates of resting state fMRI data with the increasing interest in time-dependent statistical correlations.

The paper is likely to be of broad interest to the fields of functional imaging and cognitive/behavioural neuroscience. Although the findings are novel and add to our knowledge they are not particularly surprising. But given the interest in individual differences in zero lag functional connectivity, the paper is likely to be of importance to the field.

In sum, following some further checks and balances, I would support publication in Nature Communications.

1. Since both the functional connectivity and the 1-lag matrices are $N \times N$, to first order, I guess they embody the same model complexity. However, are the degrees of freedom really the same? If the functional connectivity matrices have strong covariances between the entries, but the 1-lag entries are relatively independent, then in effect the 1-lag model is more complex – this could in theory explain why the latter explains more of the phenotypic covariance than the former – please address.
2. I'm sure the authors are aware of several prevailing controversies in the field and the paper requires more work to address these: First, was GSR performed and if so, what are the implications. If subject-specific global signal magnitude is entered as a separate parameter of the multivariate component (MC) model, does it change the primary findings of the paper.
3. Likewise, more needs to be done to address head motion. I don't think scrubbing and regression are sufficient: For example, what was regressed? Mean FWD? What is the impact of the discontinuities introduced by scrubbing on the 1-lag model? Again, as with GSR, what happens if single-subject summary motion parameters (mean FWD, maximum FWD, number of volumes scrubbed) are placed in the MC model?
4. Why did the authors use the MC model and not the approach of [1], i.e. a canonical covariates approach? That would have allowed a clearer appraisal of the 1-lag model with traditional average functional connectivity.
5. There is insufficient attention paid to the replication analyses - in what sense were they "replicated" – were they qualitatively similar or was there a more principled approach in assessing the difference in the analysis – e.g. a formal statistical comparison of the quantitative effects. Given the high dimensional models used, this cross-validation is crucial.
6. Further statistics or details are due in places – e.g. some meaning is made of the presence of a statistical difference fig 2B but not fig 2C – but is this difference itself significant? – i.e. is there an interaction effect? Similarly, line 133: "No significant difference was found" – how many tests were performed, what was the minimum p-value etc? Line 78: "FC dynamics are found to better explain scores" – please provide statistics.
7. I personally have a difference of opinion with the use of the term "dynamic" here. As the sampling frequency (TR) is much faster than the BOLD HRF, it is inevitable that a 1-lag model will have non-null elements. This does not speak very directly to any putative underlying dynamics. Moreover, I would describe a first order autoregressive time series as filtered white noise, where the filter has static parameters – i.e. a static (filtered) stochastic process. On the other hand, I think the authors have articulated a self-consistent use of the term in this and prior work, so I don't want to raise any fundamental concerns here – although a brief discussion is warranted.

Minor:

1. Are 56 items really a "deep" phenotype – compared to detailed and formal neuropsych. Testing, arguably not.
2. Line 5: "FC has been used Including Alzheimer's disease, ... and autism."
3. Line 16: "showing links to ... drug use and mind-wandering."
4. Line 169: I'm not sure a BOLD signal, even if sampled at <1 sec is an example of a "fast timescale"
5. What is the variance explained by equation (1) – i.e. what is the magnitude of the estimated coefficients to the rms noise?
6. Of the behavioural measures, how many are ordinal, binary, continuous? And are these all compatible with the assumptions of the MC model?

Reviewer #2 (Remarks to the Author):

In this manuscript, the authors report that a measure of dynamic connectivity (the coefficients of a first-order autoregressive model) captures more inter-subject variance in behavioral scores than the classically used, static, functional connectivity matrix (the data is resting-state fMRI in the Young Adult Human Connectome Project). The authors achieve this result through the use of a variance component model. They observe that the improvement afforded by their dynamic connectivity measure over the traditional static FC are more pronounced for behavioral tasks that correspond to "task performance" than for self-report scores. These are important, novel claims in the current search for neural correlates of individual differences.

As it stands, there are a few methodological points and controls that I would like to see addressed before I am convinced by the authors' conclusions. I detail them point by point below:

1) the dynamic FC estimates are obtained using the concatenated, unscrubbed data for each subject. In contrast, the static FC estimates are computed run-by-run, after carefully removing all time points that are potentially affected by motion. While I understand that scrubbing data would complicate the estimation of the AR1 model parameters, as a control it would be important to compute static FC on the same concatenated, unscrubbed data. Indeed, it could be that the dynamic FC estimates are more affected by motion artefacts, which are themselves known to be correlated with several behavioral measures (including intelligence, personality).

2) a strength of the paper is the replication of the results in an (almost) independent sample of subjects (almost, because of course they are siblings of the subjects in the discovery sample, and FC is known to have a genetic basis). However, none of the results of the replication cohort are actually shown -- a simple statement is made that "The findings shown in Figures 1-4 were replicated in a second group of 328 unrelated HCP subjects." The authors should include the results from the replication sample as supplementary figures, and provide metrics of the replicability of their main effects. This is especially important as the variance component analysis is descriptive, in-sample, rather than predictive.

3) For some measures (e.g. Cognitive Flexibility, as measured by the NIH Toolbox CardSort task), the proportion of inter-individual variance explained is about 83% for static FC, and 95% for dynamic FC. This seems extremely high -- I highly doubt that inter-individual differences on this 4-min task have 95% reliability, so even if the similarity of dynamic FC perfectly explained inter-individual differences in this task (which it is highly doubtful that they would), the expected explained variance would not exceed the reliability of inter-individual differences in the task. This observation generally calls into question the interpretability of explained variance estimates at the single score level in this multidimensional trait framework. In particular -- would the explained

variance estimates change significantly if fewer correlated behavioral measures were included in the analysis? Many of the 58 behavioral scores are inter-correlated. Perhaps the inter-correlations only contribute to the confidence on individual estimates, however it would be good to empirically confirm that the individual score results are somewhat robust to different choices of behavioral dimensions (given that the authors interpret their results at the individual score level).

4) Static and dynamic FC are passed separately through the variance component model for most results; except for Figure 4, when they are passed simultaneously to the model as two additive terms. The authors show that the combined model is superior to the independent models (though, not statistically significant). One question is, how does the explained variance from each of the terms (static and dynamic FC) compare in the combined model? In other words, when the model must partition the explained variance between dynamic and static FC, is it still the case that dynamic FC mops up more of the inter-individual variance?

Reviewer #3 (Remarks to the Author):

The manuscript "Resting brain dynamics at different timescales capture distinct aspects of human behavior" by Liégeois et al. studies the relation between static and dynamic FC and behavioral traits. They find that dynamic FC captures performance on task based measures better than static FC, and that this relation is better explained by between network FC than within network FC. This topic is highly relevant and, to my knowledge, this is the first work relating dynamical FC and behavioral traits (although there may be some around, or will be soon). The manuscript is quite clear and well referenced. I found myself, while reading the results, thinking what would I do next to dig deeper into the results, finding in the next paragraph that the authors addressed that same issue. I think it will be a very nice contribution to the field.

These are my comments:

- I find the reference to Mitra-style 'lag threads' quite confusing. The aim and methods of the lag-thread series of papers are quite different from the dynamical FC, as they provide a methodology to obtain independent, simultaneous, BOLD (not connectivity) patterns. In contrast, dynamical FC literature focuses on sequential patterns of zero-lag connectivity. I may be missing something, in any case the authors should better explain this reference in the introduction.

- Figure 1 shows that either dynamic FC explains better or there is no difference between dynamic and static. Self report measures are typically noisier and less accurate, possibly explaining the lack of significant differences. Could any of the task-related regressors be better explained by static FC? I mean, is the result trivial? After all, dynamic FC has more information than static FC to start with. Figure 3 seems to suggest that in fact it is not trivial, since for some subnetworks static FC works better than its dynamical counterpart. This might be better discussed.

- I like figure 2 very much, it is the kind of raw result one would like to visualize to have a better understanding of figure 1. Would it be possible to classify the "performance in task" regressors in finer categories?

- Figure 3, in contrast, needs some polishing. Ok, some pink bars are higher than the light blue bars and vice-versa, but error bars are too tiny to be useful. Also brain renders are not necessary. It is a lot of information, I would try to find the way to better convey it.

- Authors show that dynamic FC patterns predict performance. What are the patterns associated to high/low performance? Authors show the result of the model, but never show the actual patterns! A nice figure could be made showing the matrices associated to higher and lower performance,

both mean FC and variance. Or is the encoding in the duration of patterns?

Response to reviewers of the manuscript NCOMMS-18-14843289: *“Resting brain dynamics at different timescales capture distinct aspects of human behavior”*.

We were pleased to see a high level of enthusiasm for our work. We thank the reviewers for their close read of this manuscript and insightful comments. Several important suggestions were made for improvement of our paper. We have considered each carefully and revised accordingly. Please find below our responses (in blue) to the reviewer comments (in italics). For convenience, all changes to the manuscript are reported hereunder and quoted verbatim (normal font). Note that Figures, Tables and Equations reported in this response have an ‘R’ prefix in order to distinguish them from references to the main manuscript. All changes with respect to the original version are also underlined in the attached version of our paper. We believe this updated manuscript is much improved and hope it can now be considered for publication.

Reviewer 1

“Resting brain dynamics at different timescales,” by Liegeois et al.

This paper compares the behavioural relevance of zero lag cross correlations in resting state BOLD data to 1-lag correlations derived from a first order autoregressive model.

The study hence combines prevailing interest in the phenotypic correlates of resting state fMRI data with the increasing interest in time-dependent statistical correlations. The paper is likely to be of broad interest to the fields of functional imaging and cognitive/behavioural neuroscience. Although the findings are novel and add to our knowledge they are not particularly surprising. But given the interest in individual differences in zero lag functional connectivity, the paper is likely to be of importance to the field. In sum, following some further checks and balances, I would support publication in Nature Communications.

We thank the reviewer for the positive comments.

(Q1) Since both the functional connectivity and the 1-lag matrices are $N \times N$, to first order, I guess they embody the same model complexity. However, are the degrees of freedom really the same? If the functional connectivity matrices have strong covariances between the entries, but the 1-lag entries are relatively independent, then in effect the 1-lag model is more complex – this could in theory explain why the latter explains more of the phenotypic covariance than the former – please address.

We thank the reviewer for the careful read of our methodological framework. Indeed, when identified from an N -dimensional multivariate time series, the AR-1 model matrix and the correlation matrix do not have the same degrees of freedom. While both are of size $N \times N$, the AR-1 model matrix has in general N^2 degrees of freedom as it is non-symmetric, whereas the correlation matrix has $N \times (N - 1)/2$. However, it is important to note that the variance component model (Eq. (2) in the manuscript) takes as input a similarity matrix F that has the same degrees of freedom in both the static and dynamic cases: in both cases F is a square matrix of size N_S with $N_S \times (N_S - 1)/2$ degrees of freedom, where N_S is the number

of subjects. This is because $F(i, j)$ is -in both the static and the dynamic cases- a *scalar* that encodes similarity between connectivity patterns of subjects i and j . In other words, the difference in degrees of freedom of the static and dynamic connectivity patterns is not transferred to the static and dynamic variance component models that do have the same complexity and degrees of freedom.

To further illustrate the fact that there is no bias towards one or the other variance component model used here, consider a univariate time series $\{x_t\}$ and three competing models \mathcal{A} , \mathcal{B} and \mathcal{C} to represent $\{x_t\}$. \mathcal{A} and \mathcal{B} are AR-1 models that write $x_t = a_1 \cdot x_{t-1} + \epsilon_a$ and $x_t = b_1 \cdot x_{t-1} + \epsilon_b$, and \mathcal{C} is an AR-2 model that writes $x_t = c_1 \cdot x_{t-1} + c_2 \cdot x_{t-2} + \epsilon_c$. Let us first focus on models \mathcal{A} and \mathcal{B} : assume that in \mathcal{A} the model parameter a_1 is identified from a maximum likelihood method whereas in \mathcal{B} , the model parameter b_1 is estimated using the least squares method (e.g., Stoica and Moses, 2005; Section 4). The complexity of the computations involved to compute a_1 and b_1 are different because they use different statistics of $\{x_t\}$, but the complexity of \mathcal{A} and \mathcal{B} are the same as they are both AR-1 models. Hence there is no complexity bias giving an advantage to either \mathcal{A} or \mathcal{B} and we are only interested in which model best describes $\{x_t\}$. This is the case encountered in our main analyses (Figures 1-3): the computations to compute the static and dynamic similarity matrices from the time series are different in the two cases, but the variance component model in each case uses a single similarity matrix as input (F), with the same number of degrees of freedom. Let us now consider model \mathcal{C} . We might expect this model to better explain $\{x_t\}$ than models \mathcal{A} or \mathcal{B} because its complexity is higher. The analog of this case is our analysis of Figure 4 (dark blue bars) in which we combine static and dynamic connectivity information. In this case there is indeed an advantage of the variance component model due to increased complexity, as mentioned in the original version of our manuscript.

Finally, note that in Figure 3 of our manuscript (similarity matrices computed from subnetworks instead of the whole brain), we show that the similarity matrices derived from dynamic (AR-1) connectivity patterns explain *less* behavioral variance than the similarity matrices derived from static connectivity patterns, whereas the opposite is found at the whole-brain level (Figure 1). This also supports the fact that there is no bias towards one or the other variance component model due to complexity.

This being said, we agree with the reviewer that these considerations are important and we discuss them in the updated manuscript as follows:

Page 16 (Methods)

Complexity of static and dynamic FC variance component models - Static FC similarity matrices are built from the correlation matrix of resting-state fMRI time series. These correlation matrices are symmetric and of size $N_{ROI} \times N_{ROI}$, where N_{ROI} is the number of regions of interest, and hence their degrees of freedom (d.o.f.) is $N_{ROI} \times (N_{ROI} - 1)/2$. On the other hand, dynamic FC is built from the non-symmetric coefficient matrix of a first-order AR model of size $N_{ROI} \times N_{ROI}$, and hence their d.o.f. is N_{ROI}^2 . It should be noted, however, that this difference in d.o.f. is not present in the static and dynamic variance component models as the static and dynamic FC matrices are only used in intermediate steps of the computation of the similarity matrices of the variance component model. These similarity matrices are in both the static and dynamic case symmetric and of size N with $N \times (N - 1)/2$ degrees of freedom, where N

is the number of subjects. Therefore, there is no complexity bias giving an advantage to one or the other approach.

(Q2) I'm sure the authors are aware of several prevailing controversies in the field and the paper requires more work to address these: First, was GSR performed and if so, what are the implications. If subject-specific global signal magnitude is entered as a separate parameter of the multivariate component (MC) model, does it change the primary findings of the paper.

We did perform GSR (line 280 in Methods: *[...] and mean cortical grayordinate signal were regressed [...]*). This processing step is indeed still debated (Uddin 2017; Power et al., 2017) and we included it because the specific effect of GSR on the link between FC and behavior is studied in a separate manuscript involving 7 out of the 8 authors of the present work (Li et al., submitted). That manuscript is currently being reviewed and a **preprint** is available on BioRxiv. The main finding is that GSR strengthens the association between classical static FC measures and behavior. Therefore, as the present paper explores which behavioral information is encoded by dynamic FC *beyond* static FC, performing GSR allowed to compare dynamic FC to the 'best' static FC measures.

This being said, we agree with the reviewer that it is important to elaborate on the impact of GSR and we included two additional analyses in the revised manuscript. First, we explore the impact of including the GS amplitude as a covariate of the MC model (Figure R1B). Second, we ran the MC model using similarity matrices computed from non-GSR time series (Figure R1C). In both cases (Figure R1B-C), the main findings are reproduced. Note that we report only the main results for these additional analyses in order to avoid adding too many Figures as other new control analyses were also performed.

Page 10 (Results)

We performed a series of control analyses to evaluate the impact of various processing steps in our baseline analysis. More specifically, we tested the impact of (i) including the variance of the mean cortical grayordinate signal as a covariate in the variance component model, (ii) evaluating the static and dynamic FC matrices from fMRI time series from which the mean cortical grayordinate signal was not regressed, (iii) including head motion metrics as covariates in the variance component model, (iv) evaluating the static and dynamic FC matrices from full (i.e., uncensored) fMRI time series, (v) [...]. The variance component model appeared to be robust to these changes and in each case, our main findings were reproduced (Figures R1, S11 and S12)

Page 14 (Methods)

Mean cortical grayordinate signal regression was performed because this step was shown recently to strengthen the association between FC metrics and behavioral measures (Li et al., 2019). However, we note that not regressing mean cortical grayordinate signal yielded similar conclusions (Figure R1).

Page 35 (Supplementary Results)

We performed four control analyses to evaluate the impact of different processing steps included in our baseline analysis:

(A) - Baseline results

(B) - GS as covariate

(C) - No GS regression

(D) - Motion as covariate

(E) - No scrubbing

Figure R1: Primary findings are reproduced in different variants of the preprocessing setup. (A) Main original results. (B) Main results when including the variance of mean grayordinate signal as a covariate. (C) Main results when the static and dynamic FC matrices are computed from fMRI time series on which no mean grayordinate signal was performed. (D) Main results when including head motion metrics (mean FWD, max FWD and number of volumes scrubbed) as covariates of the variance component model. (E) Main results when the static and dynamic FC matrices are computed from full (i.e., uncensored) fMRI time series.

1. Including the variance of the mean grayordinate signal as a covariate in the variance component model (Figure R1B).
2. Computing the static and dynamic FC matrices from fMRI time series on which no mean grayordinate signal was performed (Figure R1C).
3. Including head motion metrics (mean FWD, max FWD and number of volumes scrubbed) as covariates of the variance component model (Figure R1D).
4. Computing the static and dynamic FC matrices from full (i.e., uncensored) fMRI time series (Figure R1E).

In each variant, the main results are reproduced.

(Q3) Likewise, more needs to be done to address head motion. I don't think scrubbing and regression are sufficient: For example, what was regressed? Mean FWD? What is the impact of the discontinuities introduced by scrubbing on the 1-lag model? Again, as with GSR, what happens if single-subject summary motion parameters (mean FWD, maximum FWD, number of volumes scrubbed) are placed in the MC model?

We thank the reviewer for pushing us to clarify what we did to account for head motion. First, let us note that we considered rather restrictive censoring thresholds: we censored frames with $FD > 0.2\text{mm}$ or $DVARS > 75$ whereas higher thresholds are sometimes used (e.g., Burgess et al. (2016) chose $FD > 0.39\text{mm}$ as the threshold; Caparelli et al. (2017), Thompson and Fransson (2016), and Shine et al. (2016) used $FD > 0.5\text{mm}$). Then, the HCP processing pipeline includes the regression of 24 motion-related parameters (6 classical motion parameters, their derivatives, and the squares of these 12 parameters; Salimi-Khorshidi et al. 2014). Finally, it should also be noted that the mean cortical grayordinate signal was shown to be associated with head motion (Power et al., 2014; Liu et al., 2017) and hence regressing this signal also contributes to removing global artefacts arising from motion and other physiological sources.

Regarding the impact of the discontinuities introduced by scrubbing on the estimation of the AR-1 models, we note here that the model parameters are estimated from pairs of successive time points, and hence the transitions between sections of uncensored frames are discarded from the estimation (as are the transitions between runs). Provided this is taken into account, censoring does not specifically affect AR-1 parameters estimation, the only difference being that there are less (pairs of) time points to perform the estimation. Note that this limitation also applies to the estimation of the correlation matrix, and can be expected to have a limited impact on the static and dynamic FC estimates as several runs are used in the estimation. It can be seen from Figure R1E that using uncensored fMRI time series mildly affects the estimates of explained variance and preserves our main findings.

Following the reviewer's suggestion, we also tested how including mean FD, maximum FD, and number of volumes scrubbed affected our results. It can be seen from Figure R1D that our main findings are also reproduced in this case.

We have summarized these additional control analyses and included the corresponding discussion in the updated manuscript as follows (note that additional text in the Results and Supplementary Results are not reported hereunder as they are shared with (Q2) and already reported there):

Page 14 (Methods)

[...] Both cortical and subcortical data were cleaned using the ICA-FIX method (Salimi-Khorshidi et al., 2014; Griffanti et al., 2014) and saved in CIFTI grayordinate format. This cleaning procedure included the regression of 24 motion-related parameters (6 classical motion parameters, their derivatives, and the squares of these 12 parameters). [...]

(Q4) Why did the authors use the MC model and not the approach of [1], i.e. a canonical covariates approach? That would have allowed a clearer appraisal of the 1-lag model with traditional average functional connectivity.

This is an important question and both methods have pros and cons. On the one hand, the variance components model provides an easy way to summarize high-dimensional rs-fMRI measurements and produces one scalar (variance explained) for each behavioral measure that is normalized (between 0 and 1) and directly comparable across measures and processing methods. On the other hand, CCA will likely reveal a richer pattern regarding the covariation between (and within) behavioral measures and rs-fMRI, but it would be less straightforward to compare results between static and dynamic FC because (i) we would now need to compare two eigenvalue spectra and (ii) the comparison of how static and dynamic FC explain different individual behavioral measures (as done in Figure 2) is not directly available. In addition, for the dimensionality of the data used here ($\mathcal{O}(419^2)$), CCA would require regularization or dimensionality reduction (e.g., PCA) to avoid overfitting, which complicates the analysis and adds arbitrary modeling choices.

Based on this, and since our driving research question was to explore how individual behavioral measures are explained by static vs. dynamic FC measures, we decided to use the variance component model as it seemed to be the most interpretable and natural to use in this context. This consideration has been added to the updated manuscript as follows:

Page 16 (Methods)

We used the variance component model because it provides an easy way to summarize high-dimensional FC patterns and produces one scalar (variance explained) for each behavioral measure that is normalized (between 0 and 1) and directly comparable across measures and processing methods.

(Q5) There is insufficient attention paid to the replication analyses - in what sense were they “replicated” - were they qualitatively similar or was there a more principled approach in assessing the difference in the analysis - e.g. a formal statistical comparison of the quantitative effects. Given the high dimensional models used, this cross-validation is crucial.

We apologize for not having reported the results of the replication analyses in more details. We have now included them in the Supplementary Material extensively. We also added a table reporting the p-values of the statistical tests performed in Figures 1-4 as well as the ones of the corresponding tests in the replication dataset. We have included these results as follows:

Page 10 (Results)

The findings shown in Figures 1-4 were replicated in a second group of 328 unrelated HCP subjects. More precisely, all significant differences found in Figures 1-4 were also found to be significant in the replication dataset (more details are found in Figures S5-S9 and Table R1). The replication dataset was composed of the second subject of each HCP family containing more than one person.

Table R1 reports the p -values of the statistical tests performed in Figures 1-4 (original dataset) and the corresponding tests in the replication dataset (Figures S5-S9). The p -values marked with an asterisk are the ones surviving an FDR correction at the level $q < 0.05$, when correcting for the 16 tests reported in Table R1.

Test	p-value (orig.)	p-value (repl.)
1 Mean static vs. dynamic (Fig. 1)	$8.31 \times 10^{-4*}$	$2.30 \times 10^{-3*}$
2 Static vs. dynamic in Self-Report (Fig. 2B)	2.52×10^{-1}	3.14×10^{-1}
3 Static vs. dynamic in Task (Fig. 2C)	$1.75 \times 10^{-3*}$	$2.51 \times 10^{-3*}$
4 Interaction effect (Figs. 2B&C)	$3.62 \times 10^{-3*}$	$4.30 \times 10^{-3*}$
5 Static vs. dynamic within networks (Fig. 3)	4.51×10^{-1}	3.14×10^{-1}
6 Static vs. dynamic between networks (Fig. 3)	$8.31 \times 10^{-3*}$	$2.16 \times 10^{-2*}$
7 Mean static vs. combined (Fig. 4)	$4.73 \times 10^{-4*}$	$1.15 \times 10^{-4*}$
8 Mean dynamic vs. combined (Fig. 4)	2.89×10^{-1}	3.91×10^{-1}

Table R1: List of statistical tests performed and corresponding p -values in original (orig.; 419 subjects) and replication (repl.; 328 subjects) datasets. The asterisk denotes p -values that survived FDR correction at $q < 0.05$.

(Q6) Further statistics or details are due in places – e.g. some meaning is made of the presence of a statistical difference fig 2B but not fig 2C – but is this difference itself significant? – i.e. is there an interaction effect? Similarly, line 133: “No significant difference was found” – how many tests were performed, what was the minimum p -value etc? Line 78: “FC dynamics are found to better explain scores” – please provide statistics.

We thank the reviewer for suggesting this important additional analysis about the interaction effect -also visible on Figure 2A as there are more green dots on the right- in Figures 2B and C. We find that this ‘difference of the differences’ is significantly different from zero ($p = 3.62 \times 10^{-3}$ in the original dataset). Regarding the two other suggestions, please note that all our tests were performed on the estimated *means* over all behaviors (or partitions in two groups of behaviors in Figures 2B and C), but not explicitly on individual measures in order to limit the total number of tests.

We have included the interaction effect analysis, as well as the other requested changes, in the manuscript as follows:

Pages 5, 6 and 9 (Results)

[...] We also find that the difference of the differences between static and dynamic explained variances observed in Figures 2B-C is itself different from zero ($p = 3.62 \times 10^{-3}$). This interaction effect confirms that the difference observed in Figure 2C is related to the task condition and not only driven by the main effect shown in Figure 1A. Moreover, the result of Figure 2C is reproduced using subcategories of task-based measures

(Figure S3). Overall, the better average capacity of dynamic FC to explain behavioral measures seems to be driven by its increased capacity to explain task-based measures. [...] Figure 1A shows that on average over the 58 behavioral measures, dynamic FC markers capture more behavioral variance than static FC ($p = 8.31 \times 10^{-4}$), and Figure 1B presents the results for 8 individual phenotypic measures. [...] However, no statistical difference was found between average combined results and dynamic FC ($p > 0.10$, see Table R1 for details), [...]

(Q7) I personally have a difference of opinion with the use of the term “dynamic” here. As the sampling frequency (TR) is much faster than the BOLD HRF, it is inevitable that a 1-lag model will have non-null elements. This does not speak very directly to any putative underlying dynamics. Moreover, I would describe a first order autoregressive time series as filtered white noise, where the filter has static parameters – i.e. a static (filtered) stochastic process. On the other hand, I think the authors have articulated a self-consistent use of the term in this and prior work, so I don’t want to raise any fundamental concerns here – although a brief discussion is warranted.

There are indeed several ways to present FC dynamics in the literature and we thank the reviewer for pointing out that we are using a consistent semantics to describe it. We discussed this question in more details in previous work (Liégeois et al., 2017), and would like to recall two points relating more precisely to the reviewer’s comment. First, it is technically correct to present an AR model as filtered white noise; but this is the case for all state-space models as the input of these models is usually assumed to be white noise. Hence formulating these models, which can be used for example to represent oscillatory modes, or stochastic relaxators (e.g., Neurmaier and Schneider, 2001), as filtered white noise might mask the vast repertoire of properties represented by these models. Second, it is not necessary to use time-varying parameters to represent fluctuations of a stochastic system: in Liégeois et al. (2017) we use the example of the flipping of a coin that is perfectly represented by a binomial law with ‘static’ parameters, yet the outputs of the flipping oscillates between Heads and Tails. Likewise, in the case of FC dynamics, we show in Liégeois et al. (2017) that including memory in a model with constant parameters (i.e., x_t depends on x_{t-1} as in the AR model) allows to reproduce a significant part of the temporal fluctuations observed in empirical FC time series. Such models including memory of the time series have been termed ‘dynamic’ by others than us (Theiler et al., 1992; Oppenheim and Willsky, 1997), precisely because they allow to capture a vaster repertoire of fluctuations as compared to memoryless models. Then, we agree with the reviewer that such models capture dynamics of the BOLD signal and that the link to the underlying dynamics should be further explored.

Overall, we agree that different and valid formulations could be articulated to characterize FC dynamics. We have included these considerations in our Discussion as follows:

Page 13 (Discussion)

We referred to the AR model of BOLD time series as ‘dynamic’ following the systems theory literature (Oppenheim and Willsky, 1997). This nomenclature is motivated by the fact that such models, by accounting for the memory present in the time series (i.e., x_t depends on x_{t-1}), are able to reproduce empirical fluctuations in the multi-

variate time series of interest much better than memoryless (or ‘static’, following the same nomenclature) models (Liégeois et al., 2017). The AR model is also used to compute the dominant *dynamic modes* shaping resting-state brain function (Casorso et al., 2019). Overall, this model can be seen as a compact way to summarize the temporal fluctuations of BOLD and FC time series that are directly exploited by time-varying models (Hutchison et al., 2013; Preti et al., 2016).

Minor:

(Q8) *Are 56 items really a “deep” phenotype – compared to detailed and formal neuropsych. Testing, arguably not.*

We have modified the formulation as follows:

Page 1 (Abstract)

Using resting-state fMRI and 58 phenotypic measures [...]

(Q9) *Line 5: “FC has been used Including Alzheimer’s disease, . . . and autism.”*

(Q10) *Line 16: “showing links to . . . drug use and mind-wandering.”*

We thank the reviewer for reporting these typos, which have been corrected.

(Q11) *Line 169: I’m not sure a BOLD signal, even if sampled at <1 sec is an example of a “fast timescale”*

Indeed, if compared to other modalities, the BOLD signal cannot be claimed to capture ‘fast timescales’ of brain activity. We have replaced ‘fast’ by ‘faster’ which now reflects the fact that this discussion takes place in the context of fMRI timescales.

(Q12) *What is the variance explained by equation (1) – i.e. what is the magnitude of the estimated coefficients to the rms noise?*

In the case of univariate AR models, the proportion of explained variance is usually defined as $R^2 = 1 - \text{Var}(\epsilon_t)/\text{Var}(x_t)$, where ϵ_t are the residuals and x_t are the original time series (Hamilton, 1994). The natural extension of this definition to the multivariate case is $R^2 = 1 - \|\text{Cov}(\epsilon_t)\|_F/\|\text{Cov}(x_t)\|_F$. Using this formula we find that the proportion of explained variance in the subjects of this study is $69.3\% \pm 11.2\%$. Note that it is not straightforward to express R^2 from the model parameter A in the multivariate case as from the Yule-Walker equations we have $R_0 = R_1 \cdot A + \Sigma$, where R_0 is the (co)variance of the (multivariate) time series x_t , R_1 is its 1-lagged (co)variance matrix, and Σ is the variance of the residuals ϵ_t (Stoica and Moses, 2005).

Page 15 (Methods)

The proportion of variance explained by this model, defined as $R^2 = 1 - \frac{\|\text{Cov}(\epsilon_t)\|_F}{\|\text{Cov}(x_t)\|_F}$

analogous to the definition of R^2 for univariate AR models, and where $\|\cdot\|_F$ denotes the Frobenius norm (Hamilton, 1994), is $69.3\% \pm 11.2\%$ (computed over all subjects, including the replication dataset).

(Q13) Of the behavioural measures, how many are ordinal, binary, continuous? And are these all compatible with the assumptions of the MC model?

This is important as the variance component model assumes that the behavioral measures are normally distributed. Let us first note that we performed quantile normalization of the behavioral measures, following Elliot et al., 2018. This means that each behavioral measure distribution was sorted and mapped to a linear spacing of the $]0,1[$ interval. Each behavioral measure was then replaced by the inverse normal cumulative distribution function of its mapped value, leading to a rank-preserving Gaussian redistribution of the behavioral measures.

Quantile normalization, however, should not be applied on distributions that are too 'exotic' (too skewed, too few values, etc.). To verify this, we inspected the distribution of all the behavioral measures. Among the 58 measures, none are binary, 18 are ordinal (take only integer values), 40 are continuous, and all are reasonably close to the Gaussian distribution (based on visual inspection of the histograms, and computation of skewness and kurtosis for each measure). We finally note that not performing quantile normalization did not affect our main findings.

While we already mentioned the quantile normalization operation in the manuscript, we provide further details regarding the distribution of the original behavioral measures, and on the impact of quantile normalization on the results:

Page 25 (Supplementary Methods)

Age, gender, race, education and motion (mean FD) were regressed from the 58 phenotypic measures which were then quantile normalized. To do so, each behavioral measure distribution was sorted and mapped to a linear spacing of the $]0,1[$ interval. Each behavioral measure was then replaced by the inverse normal cdf of its mapped value, leading to a rank-preserving Gaussian redistribution of the behavioral measures (Elliot et al., 2018). This normalization was motivated by the fact that Gaussianity is an assumption of the multidimensional variance component model. Quantile normalization, however, should not be applied on distributions that are too 'exotic' (too skewed, too few values, etc.). To verify this, we inspected the distribution of all the behavioral measures. Among the 58 measures, none are binary, 18 are ordinal (take only integer values), 40 are continuous, and all are reasonably close to the Gaussian distribution (based on visual inspection of the histograms, and computation of skewness and kurtosis for each measure). We finally note that results were not significantly affected if no quantile normalization was performed.

References

- Burgess G.C. et al. 2016. Evaluation of denoising strategies to address motion-correlated artifacts in resting-state functional magnetic resonance imaging data from the human connectome project. *Brain connectivity*. 6:669-80.
- Caparelli, E.C. et al. 2017. Graph theory reveals amygdala modules consistent with its anatomical subdivisions. *Scientific Reports* volume 7, Article number: 14392.
- Elliott L.T., et al. 2018. Genome-wide association studies of brain imaging phenotypes in UK Biobank. *Nature*. 562(7726):210-216.
- Hamilton, J. 1994. *Time series analysis*. Princeton, NJ: Princeton University Press.
- Li, J., Kong, R., Liegeois, R., Orban, C., Tan, Y., Sun, N., Holmes, A.J., Sabuncu, M.R., Ge, T., Yeo, B.T.T. Global Signal Regression Strengthens Association between Resting-State Functional Connectivity and Behavior. *bioRxiv* 548644; doi: <https://doi.org/10.1101/548644>.
- Liu, T.T. et al. 2017. The global signal in fMRI: Nuisance or Information? *NeuroImage*. 150:213-29.
- Neumaier, A., Schneider, T., 2001. Estimation of parameters and eigenmodes of multivariate autoregressive models. *ACM Trans. Math. Softw.* 27, 27–57.
- Oppenheim, A., Willsky, A.S., 1997. *Signals and Systems*.
- Power, J.D. et al. 2014. Methods to detect, characterize, and remove motion artifact in resting state fMRI. *Neuroimage*. 84:320-41.
- Power, J.D. et al. 2017. On global fMRI signals and simulations. *Trends in cognitive sciences*. 21(12):911-3.
- Salimi-Khorshidi, G., et al. 2014. Automatic denoising of functional mri data: combining independent component analysis and hierarchical fusion of classifiers. *Neuroimage*; 90:449–468.
- Shine J.M. et al. 2016. The Dynamics of Functional Brain Networks: Integrated Network States during Cognitive Task Performance. *Neuron*. 2016 Oct 19;92(2):544-554.
- Theiler, J., et al. 1992. Testing for nonlinearity in time series: the method of surrogate data. *Phys. D. Nonlinear Phenom.* 58 (1), 77–94.
- Thompson, W.H. and Fransson, P. 2016. Bursty properties revealed in large-scale brain networks with a point-based method for dynamic functional connectivity. *Scientific Reports* volume 6, Article number: 39156.
- Uddin L.Q. 2017. Mixed signals: on separating brain signal from noise. *Trends in cognitive sciences*. 21(6):405-6.
- Stoica, P. and Moses, R.L., 2005. *Spectral analysis of signals*. Pearson/Prentice Hall Upper Saddle River, NJ.

Reviewer 2

In this manuscript, the authors report that a measure of dynamic connectivity (the coefficients of a first-order autoregressive model) captures more inter-subject variance in behavioral scores than the classically used, static, functional connectivity matrix (the data is resting-state fMRI in the Young Adult Human Connectome Project). The authors achieve this result through the use of a variance component model. They observe that the improvement afforded by their dynamic connectivity measure over the traditional static FC are more pronounced for behavioral tasks that correspond to "task performance" than for self-report scores. These are important, novel claims in the current search for neural correlates of individual differences.

We thank the reviewer for the positive comments.

As it stands, there are a few methodological points and controls that I would like to see addressed before I am convinced by the authors' conclusions. I detail them point by point below:

(Q1) the dynamic FC estimates are obtained using the concatenated, unscrubbed data for each subject. In contrast, the static FC estimates are computed run-by-run, after carefully removing all time points that are potentially affected by motion. While I understand that scrubbing data would complicate the estimation of the AR1 model parameters, as a control it would be important to compute static FC on the same concatenated, unscrubbed data. Indeed, it could be that the dynamic FC estimates are more affected by motion artefacts, which are themselves known to be correlated with several behavioral measures (including intelligence, personality).

We apologize if this was not clear but we actually did compute the AR-1 parameter from scrubbed data. The confusion maybe comes from the fact that in the second paragraph of the Results section, we only mention that (both static and dynamic) parameters were computed "from the concatenation of the runs for each subject", without mentioning censoring. We did not refer to censoring here in order not to make the introductory paragraphs of the Results too technical. However, we specified in the Methods that both dynamic (AR-1) and static parameters were computed from uncensored timepoints. Since the AR-1 parameter is identified from pairs of successive time points, excluding censored frames is not particularly problematic as long as the transitions between different sections of uncensored frames (as well as transitions between runs) are not considered in the computation. We have further emphasized this in the updated manuscript (see hereunder).

This being said, we agree that the effect of estimating the static and dynamic parameters from uncensored data should be explored and we have included this as an additional control analysis. It can be seen from Figure R2E that the main results are not affected when estimating parameters from full (i.e., uncensored) time series.

This additional control analysis and reformulation of the methods to compute the AR-1 parameter have been included as follows:

Page 10 (Results)

We performed a series of control analyses to evaluate the impact of various processing steps in our baseline analysis. More specifically, we tested the impact of (i) [...], (iv) evaluating the static and dynamic FC matrices from full (i.e., uncensored) fMRI time series, (v) [...]. The variance component model appeared to be robust to these changes and in each case, our main findings were reproduced (Figures R2, R3 and R4).

Page 15 (Methods)

[...] The model parameter A was identified from the concatenation of the uncensored sections of the different runs, while ignoring transitions between uncensored sections and transitions between runs. [...]

Page 35 (Supplementary Results)

We performed four control analyses to evaluate the impact of different processing steps included in our baseline analysis:

1. [...]
2. [...]
3. [...]
4. Computing the static and dynamic FC matrices from full (i.e., uncensored) fMRI time series (Figure R2E).

In each variant, the main results are reproduced.

(A) - Baseline results

(B) - GS as covariate

(C) - No GS regression

(D) - Motion as covariate

(E) - No scrubbing

Figure R2: Primary findings are reproduced in different variants of the preprocessing setup. (A) Main original results. (B) Main results when including the variance of mean grayordinate signal as a covariate. (C) Main results when the static and dynamic FC matrices are computed from fMRI time series on which no mean grayordinate signal was performed. (D) Main results when including head motion metrics (mean FWD, max FWD and number of volumes scrubbed) as covariates of the variance component model. (E) Main results when the static and dynamic FC matrices are computed from full (i.e., uncensored) fMRI time series.

(Q2) a strength of the paper is the replication of the results in an (almost) independent sample of subjects (almost, because of course they are siblings of the subjects in the discovery sample, and FC is known to have a genetic basis). However, none of the results of the replication cohort are actually shown – a simple statement is made that "The findings shown in Figures 1-4 were replicated in a second group of 328 unrelated HCP subjects." The authors should include the results from the replication sample as supplementary figures, and provide metrics of the replicability of their main effects. This is especially important as the variance component analysis is descriptive, in-sample, rather than predictive.

We apologize for not having reported the results of the replication analyses in more details. We have now included them in the Supplementary Results extensively. We also added a table reporting the p-values of the statistical tests performed in Figures 1-4 as well as the ones of the corresponding tests in the replication dataset. We have included these results as follows:

Page 10 (Results)

The findings shown in Figures 1-4 were replicated in a second group of 328 unrelated HCP subjects. More precisely, all significant differences found in Figures 1-4 were also found to be significant in the replication dataset (more details are found in Figures S5-S9 and Table R2). The replication dataset was composed of the second subject of each HCP family containing more than one person.

Page 35 (Supplementary Material)

Table R2 reports the p -values of the statistical tests performed in Figures 1-4 (original dataset) and the corresponding tests in the replication dataset (Figures S5-S9). The p -values marked with an asterisk are the ones surviving an FDR correction at the level $q < 0.05$, when correcting for the 16 tests reported in Table R1.

Test	p-value (orig.)	p-value (repl.)
1 Mean static vs. dynamic (Fig. 1)	$8.31 \times 10^{-4*}$	$2.30 \times 10^{-3*}$
2 Static vs. dynamic in Self-Report (Fig. 2B)	2.52×10^{-1}	3.14×10^{-1}
3 Static vs. dynamic in Task (Fig. 2C)	$1.75 \times 10^{-3*}$	$2.51 \times 10^{-3*}$
4 Interaction effect (Figs. 2B&C)	$3.62 \times 10^{-3*}$	$4.30 \times 10^{-3*}$
5 Static vs. dynamic within networks (Fig. 3)	4.51×10^{-1}	3.14×10^{-1}
6 Static vs. dynamic between networks (Fig. 3)	$8.31 \times 10^{-3*}$	$2.16 \times 10^{-2*}$
7 Mean static vs. combined (Fig. 4)	$4.73 \times 10^{-4*}$	$1.15 \times 10^{-4*}$
8 Mean dynamic vs. combined (Fig. 4)	2.89×10^{-1}	3.91×10^{-1}

Table R2: List of statistical tests performed and corresponding p -values in original (orig.; 419 subjects) and replication (repl.; 328 subjects) datasets. The asterisk denotes p -values that survived FDR correction at $q < 0.05$.

(Q3) For some measures (e.g. Cognitive Flexibility, as measured by the NIH Toolbox Card-Sort task), the proportion of inter-individual variance explained is about 83% for static FC, and 95% for dynamic FC. This seems extremely high – I highly doubt that inter-individual differences on this 4-min task have 95% reliability, so even if the similarity of dynamic FC perfectly explained inter-individual differences in this task (which it is highly doubtful that they would), the expected explained variance would not exceed the reliability of inter-individual differences in the task. This observation generally calls into question the interpretability of explained variance estimates at the single score level in this multidimensional trait framework. In particular – would the explained variance estimates change significantly if fewer correlated behavioral measures were included in the analysis? Many of the 58 behavioral scores are inter-correlated. Perhaps the inter-correlations only contribute to the confidence

on individual estimates, however it would be good to empirically confirm that the individual score results are somewhat robust to different choices of behavioral dimensions (given that the authors interpret their results at the individual score level).

This is an important point and we thank the reviewer for pushing us to strengthen the interpretability of the the variance component model outputs. First, we want to highlight that the point estimates of the variance component model should be interpreted along with their confidence intervals (CI). For some behavioral measures, the individual error bars can be quite wide (0.2 or wider), and thus the 95% CI covers quite a wide interval and high/low variance explained can happen. For example, in the case of Cognitive Flexibility mentioned by the reviewer, the standard deviation of the dynamic estimate in the original dataset is 0.18. It can be seen in Figure S8 that in the replication dataset, the estimated static and dynamic explained variances for that behavioral measure are 74% and 81% (vs. 83% and 95% in the original dataset) which indeed suggests that estimated variance explained by static and dynamic FC in the original dataset were lying in the ‘upper’ part of the corresponding confidence interval. Then, we also tested whether the number of behavioral dimensions given as input to the variance component model could influence the estimates of the average and individual estimates of explained variances. We tested this effect both on average over different measures, and for individual measures. In the first case we randomly selected N behavioral measures and computed the mean (and standard deviation) point estimate over 100 random selections of these N behavioral measures (Figure R3A hereunder). In the second case, for each behavioral measure we ran a model including this behavioral measure plus $N - 1$ randomly chosen behavioral measures. This operation was repeated 100 times for each behavioral measure and the mean (and standard deviation) point estimates are shown in Figure R3B and C. It can be seen that there indeed seems to be an effect due to a change of this number of dimensions. However, this effect is limited and shared by static, dynamic and combined measures. We have included these additional findings and considerations in the updated manuscript as follows.

Page 10 (Results)

We performed a series of control analyses to evaluate the impact of various processing steps in our baseline analysis. More specifically, we tested the impact of (i) [...], (v) the number of behavioral measures considered in the variance component model, and (vi) [...]. The variance component model appeared to be robust to these changes and in each case, our main findings were reproduced (Figures R2, R3 and R4).

Page 37 (Supplementary Results)

We also tested the effect of the number of dimensions considered in the variance component model. We tested this effect both on average over different measures (Figure R3A), and for individual measures (Figure R3B-C). In the first case we randomly selected N behavioral measures and computed the mean (and standard deviation) point estimate over 100 random selections of these N behavioral measures. In the second case, for each behavioral measure we ran a model including this behavioral measure plus $N - 1$ other randomly chosen behavioral measures. This operation was repeated 100 times for each behavioral measure and the mean (and standard deviation) point

estimates are shown in Figure R3B-C. We finally verified that our main findings are reproduced for the special case of $N = 18$ (Figures R3D-F).

Figure R3: Impact of the number of dimensions considered in the variance component model. (A) Mean (plain) and standard deviation (dashed) of the average explained variance over N randomly chosen dimensions using 100 samples. (B-C) Mean explained variance for all individual dimensions computed using $N - 1$ other randomly chosen dimensions in the static (B) and dynamic (C) cases. (D-F) Reproduction of our main results using $N = 18$ (dashed gray line). The mean and standard deviations are computed from the 100 point estimates and not through the Jackknife approach.

(Q4) Static and dynamic FC are passed separately through the variance component model for most results; except for Figure 4, when they are passed simultaneously to the model as two additive terms. The authors show that the combined model is superior to the independent models (though, not statistically significant). One question is, how does the explained variance from each of the terms (static and dynamic FC) compare in the combined model? In other words, when the model must partition the explained variance between dynamic and static FC, is it still the case that dynamic FC mops up more of the inter-individual variance?

We thank the reviewer for the fine reading of our methodological framework. Indeed, we did not show the static and dynamic contributions *within* the combined model because it is not clear from a theoretical point of view how the shared variance is distributed to one or the other contribution. Still, we agree with the reviewer that this distribution should be checked and we show it in Figure R4 on average over the 58 behavioral measures and for 8 individual behaviors. It can be seen that as in the individual case, dynamic FC captures more behavioral variance than static FC: out of the average 45% explained by the combined model, 12% are attributed to static FC and 33% to dynamic FC (Figure R4A). As mentioned above, this result should be considered with caution as further work is required to understand how shared variance is distributed among several contributions with shared variance in a combined model.

Page 10 (Results)

We performed a series of control analyses to evaluate the impact of various processing steps in our baseline analysis. More specifically, we tested the impact of (i) [...], and (vi) the relative contributions of static and dynamic FC to the overall variance explained within the combined variance component model. The variance component model appeared to be robust to these changes and in each case, our main findings were reproduced (Figures R2, R3 and R4).

We present the relative contributions of static and dynamic FC variance in the combined model used in Figure 4. It can be seen that as in the case of individual models, dynamic FC captures more behavioral variance than static FC within the combined model: out of the average 45% explained by the combined model, 12% are attributed to static FC and 33% to dynamic FC (Figure R4A). Note that as the combined explained variance (45%) is smaller than the sum of individual static (18%) and dynamic (37%) explained variances, there is shared variance between the static and dynamic contributions. Further work is required to determine how this shared variance is distributed among various contributions from a theoretical point of view and hence this result should be considered with caution.

Figure R4: Comparison between explained variances in individual models (first two bars of each group) and distribution of variance in combined model (third bar of each group). (A) Average over 58 behavioral measures. (B) Results for 8 individual behavioral measures.

Reviewer 3

The manuscript “Resting brain dynamics at different timescales capture distinct aspects of human behavior” by Liégeois et al. studies the relation between static and dynamic FC and behavioral traits. They find that dynamic FC captures performance on task based measures better than static FC, and that this relation is better explained by between network FC than within network FC. This topic is highly relevant and, to my knowledge, this is the first work relating dynamical FC and behavioral traits (although there may be some around, or will be soon). The manuscript is quite clear and well referenced. I found myself, while reading the results, thinking what would I do next to dig deeper into the results, finding in the next paragraph that the authors addressed that same issue. I think it will be a very nice contribution to the field.

We thank Dr. Barttfeld for the positive evaluation of our manuscript.

These are my comments:

(Q1) I find the reference to Mitra-style ‘lag threads’ quite confusing. The aim and methods of the lag-thread series of papers are quite different from the dynamical FC, as they provide a methodology to obtain independent, simultaneous, BOLD (not connectivity) patterns, In contrast, dynamical FC literature focuses on sequential patterns of zero-lag connectivity. I may be missing something, in any case the authors should better explain this reference in the introduction.

The reason for including this reference was to highlight the fact that the lagged information, also exploited by the AR-1 model used here, had previously been considered. Let us just recall that we used the AR-1 model to capture dynamic FC because we showed in previous work that this model allows to efficiently summarize the temporal fluctuations of the classical zero-lagged connectivity (Liégeois et al. 2017; Figure 6). This being said, we agree with the reviewer that there are differences in the aim of the lagged-thread approaches and the AR-1 model. We have removed the first brief allusion to lag-threads in the Introduction to avoid any confusion and we now further elaborate on the differences between the two methods in the introductory part of the Results motivating the use of the AR-1 model:

Page 4 (Results)

We chose to represent FC dynamics using an AR-1 model for several reasons. First, we have shown recently that AR-1 models, by exploiting the statistical link between successive time points, capture FC dynamics significantly better than a hidden Markov model explicitly representing switches between different states with an equivalent number of parameters (Liégeois et al., 2017). Second, the hierarchical organization of brain network dynamics was found to be reproduced by an AR-1 model of fMRI time series (Vidaurre et al., 2017). Finally, lag threads, which also exploit the sequential ordering information of time series (although they focus on identifying temporal sequences of propagated activity rather than connectivity patterns) were shown to provide meaningful markers of intrinsic brain function (Mitra et al., 2015).

(Q2) Figure 1 shows that either dynamic FC explains better or there is no difference between dynamic and static. Self report measures are typically noisier and less accurate, possibly explaining the lack of significant differences. Could any of the task-related regressors be better explained by static FC? I mean, is the result trivial? After all, dynamic FC has more information than static FC to start with. Figure 3 seems to suggest that in fact it is not trivial, since for some subnetworks static FC works better than its dynamical counterpart. This might be better discussed.

This is an important point. As a preamble we would like to note that in our opinion including more statistical information in a marker does not automatically provide better/more meaningful statistical markers (e.g., is kurtosis of a distribution -which combines variance and 4th order statistical moment- *always* a better marker than its variance?). This being said, we agree that this question should be discussed and the reviewer correctly identified Figure 3 as a key result illustrating that dynamic FC does not always capture more statistical information than static FC. We have further emphasized these considerations in the updated Discussion:

Page 11 (Discussion)

Interestingly, FC dynamics within well-known resting-state networks, or between pairs of networks, did not capture more behavioral information than static FC in the same networks (Figure 3). For example, static inter-network FC was shown to explain more behavioral variance than dynamic inter-network FC on average. These results might seem counter-intuitive at first sight and suggest that similarity measures derived from *local* patterns of FC do not complement each other in the same way in the static and dynamic cases. In other words, the advantage of dynamic FC in explaining the behavioral information observed in Figure 1 is encoded in the *global* dynamic FC interaction patterns. From a methodological point of view, this also indicates that even if dynamic FC uses richer statistical information than static FC by relaxing the static assumption associated to this metric, dynamic FC should not a priori and automatically be considered as a better neuroimaging marker than static FC.

(Q3) I like figure 2 very much, it is the kind of raw result one would like to visualize to have a better understanding of figure 1. Would it be possible to classify the “performance in task” regressors in finer categories?

We thank the reviewer for this interesting suggestion. We have first tried automated methods (k-means, PCA) to determine subcategories of the 27 task-performance measures. Then, we also checked whether a partition based on the ratio between dynamic and static explained variances would be meaningful (e.g., using the ordering of green dots in Figure 2 of the manuscript). Since none of these methods seemed to yield meaningful partitions, we classified them based on the expected cognitive domains recruited by the tasks (e.g., using Barch et al., 2013), while merging under-represented categories provided they were compatible. For example, we merged ‘Social’ and ‘Memory’ measures as both categories only contained 2 measures, and also because the corresponding experiments are expected to recruit overlapping

networks such as the default mode network. The results are shown in Figure R5. It can be seen that the main finding is preserved, even if in some subcategories the difference between static and dynamic explained variance does not reach statistical significance ($p > 0.05$) which might be due to the limited number of measures composing these categories. Overall, these results strengthen our findings by showing that the better capacity of dynamic FC to explain task-based behavioral variance is reproduced in several subtypes of task-based measures. We have included these additional results in the updated manuscript as follows:

Page 6 (Results)

[...]Moreover, the result of Figure 2C is reproduced using subcategories of task-based measures (Figure R5)[...]

Page 30 (Supplementary Results)

To test whether the advantage of dynamic FC in explaining task-based behavioral measures was shared across different types of task-based measures, we computed the behavioral variance explained by dynamic and static FC in subcategories of task-based measures. These categories were determined based on the the expected cognitive domains recruited by the tasks (Barch et al., 2013). We merged ‘Social’ and ‘Memory’ measures in order to avoid under-represented categories and also because the corresponding experiments are expected to recruit overlapping networks such as the default mode network, leading to the following partitioning:

- Executive Function: Cognitive Flexibility; Fluid Intelligence; Working Memory (N-back); Working Memory (List Sorting); Relational Processing; Arithmetic; Inhibition (Flanker Task). (8)
- Emotion: Emotion Recog. – Total; Emotion Recog. – Anger; Emotion Recog. – Fear; Emotion Recog. – Happiness; Emotion Recog. – Neutral; Emotion Recog. – Sadness; Emotion face matching. (7)
- Memory/Social: Visual Episodic Memory; Verbal Episodic Memory; Social Cognition – Random; Social Cognition – Interaction. (4)
- Visuo-spatial Attention: Sustained Attention – Sens.; Sustained Attention – Spec.; Processing Speed; Spatial Orientation. (4)
- Language: Vocabulary (Pronunciation); Vocabulary (Picture Matching); Story Comprehension. (3)
- Unclassified: Dexterity. (1)

Figure R5 shows that the main finding is preserved, even if in some subcategories the difference between static and dynamic explained variance does not reach statistical significance ($p > 0.05$) which might be due to the limited number of measures composing these categories. These results strengthen our findings by showing that the better capacity of dynamic FC to explain task-based behavioral variance is reproduced in several subtypes of task-based measures.

Figure R5: Behavioral variance explained by static and dynamic FC in all ('Overall') and subcategories of task-based measures: Executive Function, Emotion, Memory/Social, Visuospatial Attention, and Language.

(Q4) Figure 3, in contrast, needs some polishing. Ok, some pink bars are higher than the light blue bars and vice-versa, but error bars are too tiny to be useful. Also brain renders are not necessary. It is a lot of information, I would try to find the way to better convey it.

We agree with the reviewer that readability of Figure 3 could be improved. We have now rearranged the diagrams and removed the y-axes labels for some of them as the scales are the same for all diagrams, leaving more space for the representation of the pairwise estimates. Then -and most importantly- we have added two panels reporting the main messages of this Figure: panel (B) shows the mean behavioral variance explained by within network static and dynamic FC, and panel C shows this for between network measures. The fact that in this latter case, static FC explains more behavioral variance than dynamic FC is a key result of the paper (cfr. Q2) and is now better highlighted.

Figure R6: (A) Behavioral variance explained by within-network (shaded diagrams) and between-network (unshaded diagrams) network static and dynamic FC. Seven cortical networks were used: visual (VIS), somatomotor (SM), dorsal attention (D-Att), salience (Sal), limbic (Lim), frontoparietal (FP), default mode network (DMN) and we also gathered the 19 subcortical areas (Sub). (B) There is no statistically significant difference in behavioral variance explained by within-network static and dynamic FC. (C) Between-network static FC explains more behavioral variance than between-network dynamic FC ($p = 8.31 \times 10^{-3}$).

(Q5) Authors show that dynamic FC patterns predict performance. What are the patterns associated to high/low performance? Authors show the result of the model, but never show the actual patterns! A nice figure could be made showing the matrices associated to higher and lower performance, both mean FC and variance. Or is the encoding in the duration of patterns?

We first note that the association model used here is multivariate in the sense that FC similarity between subjects is computed using all the entries of the FC matrices, and summarized by one scalar for each pair of subjects in the similarity matrix F defined in Eq. (2) of the paper. This presents the advantage of characterizing FC interactions as a whole, allowing to compare subjects based on their global FC patterns (Results of Figure 1 and 3 illustrate the importance of considering these interactions as a whole as opposed to separately). The corresponding drawback is that the model does not explicitly link each FC connection to

behavioral measures. Still, we agree with the reviewer that including such a pattern of ROIs interactions driving the association between dFC and performance would significantly improve our manuscript. We have therefore further explored how this could be done and found a reformulation of the variance component model that allows to evaluate the relative contribution of each ROI x ROI interaction to the overall explained variance. This connectivity pattern is represented in new Figure 5 of the Results and it can be seen that specific subnetworks of the default and control networks are mainly contributing to the overall explained variance.

The methodological details, additional results and discussion involving this new analysis are reported hereunder. We believe this important new piece of result nicely completes the overall contribution of the paper and we thank the reviewer for having pushed us to explore this possibility.

Page 9 (Results)

We now explore which dynamic FC interactions contribute to the overall association with task performance (Figure 2C). We used a reformulation of the variance component model defined in Eq. (2) that revealed the relative contribution of the interaction between each pair of (sub)networks to the overall explained variance (Methods Eq. (S15)). The results are shown in Figure R7. It can be seen that default C and frontoparietal C, together with the subcortical regions, are contributing the most to the association between dynamic FC and task performance.

Figure R7: Dynamic FC interactions contributing the most to the association with task performance. Networks and corresponding colors are the same as in Figure R6, and subnetworks are defined following the 17-network parcellation of Schaefer et al. (2017), as reported in Figure R8. The colors of the edges are defined by their destination and only connections surviving an FDR correction at the level $q = 0.05$ are shown.

Page 11 (Discussion)

In Figure 2, we show that FC dynamics specifically encode measures of performance in a task, such as working memory tasks, whereas static and dynamic FC explain self-reported measures, such as the perception of loneliness, equally well. This additional information is found to be encoded in the global dynamic FC patterns, and not confined to single areas or networks (Figure 3). More precisely, Figure R7 suggests that the default mode and frontoparietal networks drive the integration of the dynamic FC coming from other networks. This is in line with previous findings identifying these areas as hubs of the dynamic functional connectome (Schaefer et al., 2014), and further supports the importance of coupled default network and frontoparietal activities during task performance (Spreng et al., 2010). Taken together, [...]

Page 25 (Supplementary Methods)

The contribution of pairwise interactions to the overall explained variance is obtained from the following model:

$$y = Wu + e, \tag{R1}$$

where y is a vector encoding one behavioral measure for the N subjects, u is a random-effects vector of length P , the number of entries in the FC matrices, W is an $N \times P$ matrix with centered and unit-variance lines, and e is the normally distributed residual with variance σ_e . Assuming each element of u is independent and follows a normal distribution with variance σ_c/P , then the model can be turned into the variance component model of Eq. (2) we used: $Cov(y) = \sigma_c \cdot F + \sigma_e \cdot I$, where $F = W \cdot W^T/P$ is the similarity matrix of the connectome between pairs of individuals. Then, using the best linear unbiased predictor of u following Yang et al. (2011), the entries of u^2 provide a scaled estimate of the variance explained by each pair of ROIs. To evaluate the contributions of ROI pairs over all behavioral measures of task performance, u^2 was computed for each task-related behavioral measure and weighted by the loadings of the first principal component of these behavioral measures. This weighted u^2 was used to produce Figure R7, as further detailed in the Supplementary Results (Figure R8).

Figure R8 shows the pairs of ROIs of the dynamic FC pattern most contributing to the task-behavioral explained variance, obtained from Eq. (R1). This result is used to generate Figure R7 by averaging contributions in the network-interactions using the 17-network parcellation (Schaefer et al., 2017). To explore whether the explained variance was concentrated in some network interactions, we compared these values to the ones obtained after randomly shuffling lines and columns in Figure R8. The interactions shown in Figure R7 are the ones surviving an FDR correction at the level $q = 0.05$.

Figure R8: *(Left)* Contribution of each pairwise ROI interaction to task-behavioral explained variance, following Eq. (R1). The color code corresponds to the 7-network parcellation used in Figures 3 and 5. *(Right)* Correspondance with 17-network sub-parcellation (Schaefer et al., 2017).

Pablo Barttfeld

REVIEWERS' COMMENTS:

Reviewer #1 (Remarks to the Author):

I thank the authors for their clear and detailed responses to my prior concerns. I have no further reservations.

Reviewer #2 (Remarks to the Author):

The authors have addressed the concerns I raised in the first round of reviews, and I am happy to recommend the manuscript for publication.

A minor note: I would appreciate if the authors could highlight that their replication sample is not completely independent from their discovery sample -- all subjects in the replication sample are siblings of subjects in the discovery sample.

Reviewer #3 (Remarks to the Author):

All my concerns have been addressed

Response to Reviewers:

Please make the minor changes suggested by Ref #2

"A minor note: I would appreciate if the authors could highlight that their replication sample is not completely independent from their discovery sample -- all subjects in the replication sample are siblings of subjects in the discovery sample."

We have included this in the 'Replication Data set' subsection as follows: 'The replication dataset was composed of the second subject of each HCP family containing more than one person. We note that it is therefore not completely independent from the discovery dataset.'